# Transmission dynamics and baseline epidemiological parameter estimates of Coronavirus disease 2019 pre-vaccination: Davao City, Philippines

**Loreniel E. Añonuevo**[1,2], **Zython Paul T. Lachica**[1,3,4¤], **Deza A. Amistas**[1], **Jayve Iay E. Lato**[1], **Hanna Lyka C. Bontilao**[1], **Jolly Mae G. Catalan**[5], **Rachel Joy F. Pasion**[5], **Annabelle P. Yumang**[5], **Alexis Erich S. Almocera**[1,6], **Jayrold P. Arcede**[1,2], **May Anne E. Mata**[1,3,4,6]*, **Aurelio A. de los Reyes V**[1,4,7,8]

**1** Center for Applied Modeling Data, Analytics, and Bioinformatics for Decision Support Systems in Health, University of the Philippines Mindanao, Davao City, Philippines, **2** Department of Mathematics, Caraga State University, Butuan City, Philippines, **3** Interdisciplinary Applied Modeling (IAM) Laboratory, University of the Philippines Mindanao, Davao City, Philippines, **4** University of the Philippines Resilience Institute, University of the Philippines Diliman, Quezon City, Philippines, **5** Department of Health – Davao Center for Health Development, Davao City, Philippines, **6** Department of Mathematics, Physics, and Computer Science, University of the Philippines Mindanao, Davao City, Philippines, **7** Institute of Mathematics, University of the Philippines Diliman, Quezon City, Philippines, **8** Biomedical Mathematics Group, Pioneer Research Center for Mathematical and Computational Sciences, Institute for Basic Science, Daejeon, Republic of Korea

¤ Current address: Nuffield Department of Medicine (OX3 7BN) & St. Cross College, University of Oxford, Oxford, United Kingdom
* memata@up.edu.ph

**Data Availability Statement:** The data contain potentially sensitive information. The data underlying the results presented in the study are

## Abstract

The Coronavirus disease 2019 (COVID-19) has exposed many systemic vulnerabilities in many countries' health system, disaster preparedness, and adequate response capabilities. With the early lack of data and information about the virus and the many differing local-specific factors contributing to its transmission, managing its spread had been challenging. The current work presents a modified Susceptible-Exposed-Infectious-Recovered compartmental model incorporating intervention protocols during different community quarantine periods. The COVID-19 reported cases before the vaccine rollout in Davao City, Philippines, are utilized to obtain baseline values for key epidemiologic model parameters. The probable secondary infections (i.e., time-varying reproduction number) among other epidemiological indicators were computed. Results show that the cases in Davao City were driven by the transmission rates, positivity proportion, latency period, and the number of severely symptomatic patients. This paper provides qualitative insights into the transmission dynamics of COVID-19 along with the government's implemented intervention protocols. Furthermore, this modeling framework could be used for decision support, policy making, and system development for the current and future pandemics.

available from the Department of Health-Davao Center for Health Development. One can email them using this email address: doh11davao@gmail.com.

**Funding:** This research is funded by the Department of Science and Technology-Philippine Council for Health Research and Development through its Niche Center in the Regions for Research and Development, "Center for Applied Modeling, Data Analytics, and Bioinformatics for Decision Support Systems in Health".

**Competing interests:** The authors have declared that no competing interests exist.

## Introduction

The Coronavirus disease 2019 (COVID-19) caused by SARS-CoV-2 virus has globally spread since its first detection in Wuhan, China, last December 2019 and has caused millions of deaths worldwide [1–4]. From then on, different control measures have been implemented to mitigate its spread. In the absence of vaccines, different countries have resulted in the implementation of non-pharmaceutical interventions (NPIs) [5–10], such as the quarantine measures/isolation, contact tracing, physical distancing, and mass testing. In the Philippines, various levels of community quarantines have been implemented in different parts of the country since the first confirmed COVID-19 case was reported [11]. These community quarantines are classified as enhanced community quarantine (ECQ), modified enhanced community quarantine (MECQ), general community quarantine (GCQ), or modified general community quarantine (MGCQ). Regardless of the quarantine classification, the general public is urged to stay in their respective households, practice physical distancing, and wear face masks and face shields to lessen the spread of the disease. School closures, halting of mass gatherings and other potentially superspreading events (e.g., religious activities) have been imposed. NPIs are continually being implemented to prevent a surge of cases due to the crowding of individuals [12, 13]. Noteworthy, however, is the apparent difference in the strictness in implementing different control measures between quarantine classifications as summarized in Table 1 [14]. Furthermore, the number of deployed quarantine enforcers and the number of monitoring stations put up differs among respective quarantine classifications. Even between regions in the Philippines under similar quarantine classification, differences in resources and workforce, among others, contribute to variability in the efficacy of the intervention. Hence, assessing the effects of the different quarantine classifications on a regional, provincial, or city-level could help decision-makers formulate a more appropriate policy to mitigate the health crisis.

A baseline measure should be developed to properly comprehend the effect of an intervention to the COVID-19 cases; this will be a basis for determining any subsequent changes brought about by these interventions [15, 16]. Epidemiological models help identify critical factors of non-pharmaceutical interventions (NPIs) implementation, such as timing and frequency, that support the control of disease spread [17]. Understanding the parameters that determine the course of an epidemic is critical for health-related decision-making, as it allows for the development of disease mitigation and control methods, as well as the provision of treatment to individuals who have been infected or become ill [18]. These models have been used as guides to policy and decision-makers as well as implementers to combat outbreaks [19–21]. To this end, several mathematical models on the dynamics of COVID-19 with NPIs have been developed and published to project the different COVID-19 transmission scenarios

**Table 1. Quarantine classification, sample restrictions, Davao City, Philippines [14].**

|  | Quarantine Classification | | | |
|---|---|---|---|---|
|  | **ECQ** | **MECQ** | **GCQ** | **MGCQ** |
| Population | 100% stay at home | 100% stay at home | Elderly & youth | Permissive socio-economic activities with minimal public health standards |
| Gathering | Not Allowed | Restricted to 5 max | Restricted to 10 max | |
| Travel | No public transport; no domestic flights; limited international flights | No public transport; limited international flights; controlled inbound travels | Public transport with strict distancing | |
| Government | Skeletal onsite; Others work from home | Skeletal onsite; Others work from home | Alternative work arrangement; 4-day workweek | |

on a larger scale, i.e., national level [22, 23]. Nevertheless, assessing the effects of different quarantine classifications to key epidemiological parameters on a more specific scale, i.e., regional level, has not been widely documented.

The high transmissibility and virulence of SARS-CoV-2 resulted in a significant number of severe and critical cases requiring specialized treatment and intensive care beds— forcing the development of predictive models capable of estimating healthcare demands and assisting decision-making [24–26]. For better contextualization of the model, local COVID-19 dynamics and NPIs in Davao City were used in the current case study. Davao City is the largest city in the Philippines based on land area. It is located in Region 11 (Davao Region), the most populous region in Mindanao—the southern part of the country [27]. The statistics from the Department of Health-Davao Center for Health Development (DOH-DCHD) show that Davao City accounts for almost 57% of Davao Region's COVID-19 cases (as of June 26, 2021). Davao City is also the capital of Region 11, serving as the focal business hub and other activities in the region. Hence, the importation and transmission risk of COVID-19 from and to neighboring provinces and regions is high.

This paper modified the classic Susceptible-Exposed-Infected-Recovered ($SEIR$) deterministic compartmental model [28] into a Susceptible-Exposed-Infectious Hospitalized-Infectious Monitored-Recovered ($SEI_HI_MR$) model to describe the dynamics of COVID-19 in Davao City incorporating the Philippines' Department of Health's quarantine and isolation protocols, namely quarantining exposed individuals and isolating confirmed positive cases according to the severity of symptoms or presence of comorbidity [13]. This model incorporated an outflow from the $E$ compartment back to $S$ similar to [29]— a dynamic that has not been widely applied to models in a Philippine regional setting. This link signified that, instead of the standard classification of the $E$ compartment [28], not all exposed individuals in this model are on a latent stage of the disease, do not become infectious, and hence become susceptible again after they were released from quarantine or isolation. Moreover, the model also incorporates the progression or worsening of symptoms of some previously asymptomatic positive cases (*i.e.,* presymptomatic). Disease-related deaths also in this model are assumed to only occur among severely symptomatic patients. Furthermore, in Davao City's context, isolation and, or quarantining of the infectious individuals do not guarantee that the patient has stopped contributing to the pathogen's transmission; viral spread may occur before an intervention due to delays in test result. Moreover, health and emergency workers can still get infection from quarantined or isolated individuals. Hence in this model, we assumed that the compartments $I_H$ and $I_M$ equally drive the transmission regardless of isolation or quarantine. Nevertheless, the contact tracing method in Davao City during the period covered in this paper is assumed to be efficient whereby minimizing the possibility of an untraced infectious individual. Thus, the model does not incorporate a compartment for undetected cases to make analysis tractable.

We aimed to estimate the baseline epidemiological parameters, i.e., pre-vaccination COVID-19 period, using the least-square method and bootstrapping techniques in quantifying parameter uncertainty. Epidemiological measures such as the basic reproduction number($R_0$), statistical time-varying reproduction number ($R_t^s$), and the deterministic time-varying reproduction number ($R_t^d$) have also been computed. These numbers generate insightful thresholds on the secondary infections generated by a COVID-19 infectious patient upon interaction with the susceptible population [30]. We also provided basic local stability analysis of the model. The COVID-19 outbreak has led researchers to investigate numerous factors of disease transmission and evolution [31, 32]. The empirical results from the generated model can guide the local government unit in reviewing their respective protocols to mitigate the spread of

COVID-19. Furthermore, the model described here is general enough to be used in studying COVID-19 dynamics in other regions, cities, or municipalities.

## Materials and methods

### Ethics statement

In compliance with the Joint Memorandum Circular No. 2020–0002 by the Philippines' Department of Health on the data privacy guideline, processing and disclosure of COVID-19-related data for disease surveillance and response, the study was reviewed by the Department of Health XI Cluster Ethics Review Committee with protocol number P211111601. Furthermore, this study does not involve human participants or personally identifiable information.

### Model formulation

In this paper, the closed homogeneous population was divided into five population classes, namely, susceptible ($S$), quarantined/exposed ($E$), infectious individuals treated in either a monitoring facility ($I_M$) or hospitals handling severe cases ($I_H$), and recovered ($R$) individuals. We specifically defined the compartments as follows:

- Susceptible ($S$): Individuals who have not been infected with COVID-19 but are at risk of contracting the disease;

- Exposed ($E$): Individuals who were under quarantine who may have been exposed to the disease and were still in the latent stage or who may have been exposed but have not necessarily contracted the virus;

- Hospitalized ($I_H$): Infected individuals confirmed positive for COVID-19 treated in COVID-19-designated referral hospitals and exhibited moderate to critical symptoms or have known comorbidities;

- Monitored ($I_M$): Infected individuals confirmed positive for COVID-19 under monitoring in designated Temporary Treatment and Monitoring Facilities (TTMFs). These are infected persons who exhibited no symptoms (asymptomatic) or who had mild COVID-19 symptoms; and

- Recovered ($R$): Individuals who recovered from the disease and possessed a certain level of immunity.

The schematic diagram in Fig 1 depicted the disease transmission dynamics, with the corresponding parameter descriptions, range of values, and sources, were detailed in Table 2. Herein, the mean residence time in $E$, in $I_M$, and in $I_H$ are respectively given as

$$A^{-1} = (\psi(1 - r) + \delta_H rq + \delta_M r(1 - q) + \delta)^{-1},$$
$$B^{-1} = (\varphi_H m + \gamma_M(1 - m) + \delta)^{-1}, \text{ and}$$
$$C^{-1} = (\mu_H + \delta + \gamma_H)^{-1}.$$

The system of differential equations that represents the transmission dynamics of the SARS-CoV-2 virus in the city was obtained by adding the inflow rate and subtracting the outflow rate for each respective component. The following system (1) mathematically described

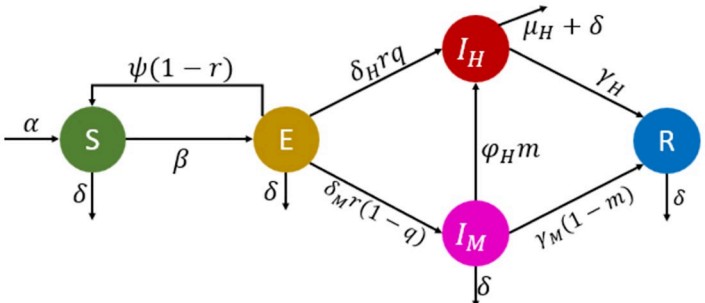

**Fig 1. Schematic diagram of the regional COVID-19 transmission dynamics.** Circles represent the compartments (state variables of the model), while arrows indicate flows between compartments $SEI_HI_MR$. The outflow from $I_H$ indicated by $\mu_H$ indicates disease-induced mortality.

the disease dynamics

$$
\begin{aligned}
\frac{dS}{dt} &= \alpha - \delta S - \frac{\beta(I_M + I_H)S}{N} + \psi(1-r)E, \\
\frac{dE}{dt} &= \frac{\beta(I_M + I_H)S}{N} - (\psi(1-r) + \delta_H rq + \delta_M r(1-q) + \delta)E, \\
\frac{dI_H}{dt} &= \delta_H rqE + \varphi_H mI_M - (\mu_H + \delta + \gamma_H)I_H, \\
\frac{dI_M}{dt} &= \delta_M r(1-q)E - (\varphi_H m + \gamma_M(1-m) + \delta)I_M, \\
\frac{dR}{dt} &= \gamma_H I_H + \gamma_M(1-m)I_M - \delta R, \\
\frac{dN}{dt} &= \alpha - \delta N - \mu_H I_H
\end{aligned}
\tag{1}
$$

**Table 2. Parameter description, values, and sources for Davao City, Philippines.**

| Parameter | Description | Value/Range | Unit | Source |
|---|---|---|---|---|
| $N$ | Total Population | 1,816,987 | indv | [33] |
| $\beta$ | Transmission rate | [0, 1] | per day | Estimated |
| $r$ | Positivity proportion of Exposed that becomes infectious | [0, 1] | % per day | Estimated |
| $\delta_H$ | $E \rightarrow I_H$: Reciprocal of the latency period from onset of illness to Hospital Admission | [0, 1] | per day | * |
| $\delta_M$ | $E \rightarrow I_M$: Reciprocal of the latency period from onset of illness to TTMF Admission | $\frac{1}{3}$,[0, 1] | per day | * |
| $\gamma_H$ | $I_H \rightarrow R$: Recovery rate of $I_H$ individuals | [0, 1] | per day | * |
| $\gamma_M$ | $I_M \rightarrow R$: Recovery rate of $I_M$ individuals | [0, 1] | per day | * |
| $\mu_H$ | $I_H \rightarrow D$: COVID-19 related death rate | [0, 1] | per day | * |
| $q$ | Severity proportion of reported cases that are moderate-critical | [0, 1] | % per day | * |
| $m$ | Presymptomatic proportion of $I_M$ cases that become critical/severe | [0, 1] | % per day | * |
| $\psi$ | Discharging rate of Exposed | $\frac{1}{14}$ | per day | [13] |
| $\varphi_H$ | $I_M \rightarrow I_H$: Hospitalization rate of $I_M$ | $\frac{1}{5}$ | per day | * |
| $\delta$ | Natural Death Rate | $\frac{1}{26017.2}$ | per day | [34] |
| $\alpha$ | Recruitment Rate | 69.8379** | per day | [34] |
| $A^{-1}$ | Mean residence time in $E$ | [0, 20] | days | Calculated |
| $B^{-1}$ | Mean residence time in $I_M$ | [0, 20] | days | Calculated |
| $C^{-1}$ | Mean residence time in $I_H$ | [0, 20] | days | Calculated |

* estimated from the data,

**$\alpha \approx N\delta$

with/where $S(t)$, $E(t)$, $I_M(t)$, $I_H(t)$, and $R(t)$ are nonegative, and $N(t) = S(t) + E(t) + I_M(t) + I_H(t) + R(t)$. The parameters were piecewise functions of time subject to the duration of implementation of the different levels of quarantine classifications (i.e., ECQ, MECQ, GCQ, and MGCQ). It could be easily seen from the system (1) that because $\frac{dN}{dt} = \alpha - \delta N - \mu_H I_H$, $sup_{t \to +\infty} N(t) \leq \frac{\alpha}{\delta}$ when $\frac{dN}{dt} = 0$. As such, the system's feasible region is

$$\Omega = \{S, E, I_M, I_H, R \in R_+^5 : 0 \leq N \leq \frac{\alpha}{\delta}\}.$$

Hence, the model is well posed, and all solutions remain in $\Omega$.

## Data

Publicly available data on the daily COVID-19 new infected cases in the Philippines can be accessed through the government data drop [35]. However, the data used in this study were obtained under a partnership and non-disclosure agreement from the DOH—DCHD: Regional Epidemiology and Surveillance Unit (RESU). This partnership and data-sharing agreement were sought for better and more efficient data, and result validation, which would have been difficult if the data was acquired from the data drop. The epidemiological dataset used included dates of illness' onset, dates of surveillance report, and the health status upon admission of confirmed cases (e.g., asymptomatic, mild, moderate, severe, and critical). The dataset was between March 8, 2020, to March 5, 2021. We categorized the epidemic data according to two groups: the infected individuals under monitoring in TTMFs (e.g., asymptomatic and mild cases), and the infected patients admitted to COVID-19 referral hospitals (e.g., moderate, severe, and critical cases). Even though the working model is a baseline model that does not distinguish vaccination as a disease control measure, the model could be used to assess pre-vaccination transmission dynamics. Hence, we only used data before the start of the vaccine rollout in Davao City, which was on March 5, 2021. For reproducibility, a sample dataset which is in compliance with the non-disclosure agreement could be accessed through [36].

## Community quarantine timeline pre-vaccination

Pre-vaccination, the city government of Davao, following the recommendations and guidelines issued by the Inter-Agency Task Force (IATF), imposed NPIs such as social distancing, lockdowns, curfews, and office and school closures, among others, to reduce disease transmission. The intensity of the implementation of NPIs varied according to the type of community quarantine imposed as summarized in Table 1. Thus, we divided the epidemiological data into four distinct periods according to the intensity and classification of community quarantine implemented in the city. The government first imposed community quarantine (CQ) in Davao City on March 15—April 3, 2020. On April 4—May 15, 2020, Davao City was placed under ECQ. For simplicity, the study assumes that Davao City is under ECQ from March 8, 2020, to May 15, 2020. The first quarantine period is referred to as Q1. The periods Q2, Q3, and Q4 occurred on May 16—June 30, July 1—November 19, and November 20—March 5, 2021, which were under GCQ, MGCQ, and back to GCQ, respectively.

## Sensitivity analysis

Sensitivity analysis was performed *a priori* to parameter estimation. This was used as the basis why certain parameters needed to be identified while some were fixed. Parameters with higher sensitivity needed to be estimated reliably subject to the available information. We performed

uncertainty and sensitivity analysis using the Latin Hypercube Sampling (LHS) and Partial Rank Correlation Coefficient (PRCC) method [37]. To perform the LHS analysis, 11 of the 13 system parameters plus a dummy were varied simultaneously while we fixed the values of the $\alpha$ and $\delta$ parameters according to Table 2. We assumed a uniform distribution for the 11 parameters with [0, 1] bounds for the transmission rate $\beta$ and [0, 1] bounds for the rest of the parameters. The LHS/PRCC method was then applied utilizing the codes by Massey, which were publicly available on GitHub [38]. The method involved generating 10,000 samples via the LHS scheme to populate the LHS matrix. Each row on the LHS matrix was then passed to the $SEI_H I_M R$ model to run 10,000 Monte Carlo simulations. We considered three (3) outcome measures: the infectious monitored individuals ($I_M$), hospitalized infectious individuals ($I_H$), and the deterministic time-varying reproduction number ($R_t^d$). Parameters with PRCC values close to −1 and +1 were highly negatively (positively) correlated to the selected outcome measures.

## Model parameterization

Model parameters were estimated following the methods of Chowell, and Banks et al. [39, 40]. The model was calibrated to the daily new infected cases in Davao City from March 8, 2020 (the first case of COVID-19) to March 5, 2021. The model calibration was performed individually for each quarantine period considered (e.g., Q1—Q4). The model has 13 epidemiological parameters. Two (2) of these parameters were estimated: the transmission rate ($\beta$), and the positivity proportion ($r$). First, a seven (7)-day moving average filter was applied to smoothen the random variations in the daily incidence data. To fit the asymptomatic-or-mild daily incidence data to the model, we defined the newly infected people under monitoring as $M(t) = \delta_M r(1 − q)E$. The moderate-to-critical daily incidence data were fitted to the model using the new hospitalized infected people defined as $H(t) = \delta_H rqE$. The data-fitting problem was then solved using the least squares (LS) technique given by the objective function $f t_i) = argmin \Sigma_{i=1}^{n} (f(t_i) − y(t_i))^2$, $y(t_i)$ was the observed daily new cases, and $f(t_i)$ was the corresponding model simulation. In particular, $f(t_i) = (M(t_i), H(t_i))$ and $y(t_i)$'s are the daily reported monitored and hospitalized cases. The fitting procedure was executed using the MATLAB `lscurvefit` function with numerical optimization through a trust-region reflective algorithm [41]. Other parameter values were either estimated from the data [42, 43] or taken from existing published literature. The complete list of the system parameters and their sources is presented in Table 2. The estimated parameter values are presented in Table 3. These values were then analyzed for feasibility and were verified or compared to other existing literature. The analysis is expounded in the results section.

**Table 3. Estimated parameter values per quarantine period for Davao City, Philippines.**

| Parameter | Q1 (ECQ) | Q2 (GCQ) | Q3 (MGCQ) | Q4 (GCQ) |
|---|---|---|---|---|
| $\beta$ | 0.2862 | 0.1284 | 0.2359 | 0.2142 |
| $r$ | 0.1013 | 0.1758 | 0.1702 | 0.1238 |
| $\delta_H$ | $\frac{1}{7}$ | $\frac{1}{5}$ | $\frac{1}{3}$ | $\frac{1}{3}$ |
| $\gamma_H$ | 0.0238 | 0.0257 | 0.0244 | 0.0193 |
| $\gamma_M$ | $\frac{1}{19}$ | $\frac{1}{16}$ | $\frac{1}{14}$ | $\frac{1}{11}$ |
| $\mu_H$ | 0.1223 | 0.1579 | 0.2928 | 0.3002 |
| $q$ | 0.2271 | 0.0788 | 0.0842 | 0.0742 |
| $m$ | 0 | 0 | 0.0002 | 0.0054 |

## Bootstrapping method

The bootstrapping method was used to simulate the lower and upper bounds (95% confidence level) around the model fit in assessing parameter identifiability [39]. The parametric bootstrapping approach generated 10,000 bootstrap realizations assuming a Poisson error structure of the best fit from the LS fitting method. The generated bootstrap realizations were used to re-simulate the $SEI_HI_MR$ model and to derive the empirical distributions of the estimated parameters within a 95% confidence interval.

## Time-varying reproduction number

The daily statistical time-varying reproduction number ($R_t^s$) in this paper was computed following the methods of Cori and colleagues with prior mean $\mu_{SI} = 2.6$ and standard deviation $\sigma_{SI} = 2$ [37, 44] and mean $\mu_{SI} = 4.8$ and standard deviation $\sigma_{SI} = 2.3$ [45]. Furthermore, the COVID-19 cases according to the onset of symptoms, were used to statistically estimate the $R_t^s$. On the other hand, a formula for the deterministic time-varying reproduction number ($R_t^d$) was also derived from the compartmental model using the next-generation matrix method [30]. The $R_t^s$ was then compared to the $R_t^d$.

## Results and discussion

### Qualitative analyses

Some of the qualitative analyses performed in this study involved computing the basic ($R_0$) and time-varying deterministic reproduction numbers ($R_t^d$), and solving for the model's local stability, equilibrium points, and elasticity indices. The detailed steps to these analyses, proof and calculation can be found in the supplementary S1 File. The basic reproduction number ($R_0$) was the initial reproduction number of the pathogen at the start of the pandemic, whereas the deterministic time-varying reproduction number $R_t^d$ was the reproduction number at any particular point in time $t$. The $R_t^d$ described the average number of secondary infections caused by an infective individual in a susceptible population in the presence of control measures [38]. It provided a threshold for disease outbreak: if $R_t^d < 1$, the disease cannot persist; when $R_t^d \geq 1$, the virus can spread within the population where each infectious individual produced at least one new infection [46, 47]. The derived closed-form formulas of $R_0$ and $R_t^d$ were as follows:

$$R_0 = \beta r \frac{\delta_M(1-q)}{AB} + \beta r \frac{\delta_H qB + \varphi_H m \delta_M(1-q)}{ABC} > 0,$$

$$R_t^d = \beta r \frac{\delta_M(1-q)S}{ABN} + \beta r S \frac{\delta_H qB + \varphi_H m \delta_M(1-q)}{ABCN},$$

$r, q \in [0, 1]$ where $A = \psi(1-r) + \delta_H rq + \delta_M r(1-q) + \delta > 0$, $B = \varphi_H m + \gamma_M(1-m) + \delta > 0$, and $C = \mu_H + \delta + \gamma_H > 0$. Recall that $A^{-1}$, $B^{-1}$, and $C^{-1}$ are just the mean residence time of compartments $E$, $I_M$, and $I_H$, respectively. This result meant that the average number of secondary infections caused by an infective individual is directly proportional to the transmission rate $\beta$ and the positivity proportion $r$ and is inversely proportional to $A$, and $B$.

To further determine how the parameters affect $R_0$, we solved for the elasticity index (normalized sensitivity index) [48]. The elasticity index measures the relative change of $R_0$ with

respect to a parameter, for example, $\beta$, denoted by $\Upsilon_{\beta}^{R_0}$, such that

$$\Upsilon_{\beta}^{R_0} = \frac{\partial R_0}{\partial \beta} \times \frac{\beta}{R_0}.$$

The following are the proportionality and the elasticity indices for the relevant parameters:

1. The direct proportionality of the transmission rate $\beta$ to $R_0$ is supported by its elasticity index of 1. This means that a unit increase of $\beta$ results in a unit increase of $R_0$. Hence, any increase or decrease of the transmission of the pathogen from one person to another directly affects the size of the pandemic and its mitigation. Conversely, this result supports that the measures designed to hamper this transmission, such as wearing facemasks, hand washing, disinfection, observing physical distancing, and isolation, among others, [49, 50] can directly slow down the spread of the disease.

2. The direct proportionality of the positivity proportion $r$ to $R_0$ is also quantified with an elasticity index of 1. This means that a unit increase in $r$ also results in a unit increase of $R_0$. Thus, interventions designed to decrease $r$ could directly lead to the mitigation of the spread of the disease. The interventions related to $r$ are, but not limited to, contact tracing and testing. An increase in the effort in these interventions will directly increase the identified positive cases, thereby helps curb the pandemic for better management. That is, the more contact traced, quarantined, and tested individuals, the better, regardless if only a few among them become a confirmed positive case. It is even more ideal to have a lower positivity rate among tested individuals.

3. The elasticity index of $A$ on $R_0$ is $-1$. This backs the inverse proportionality of $A$ to $R_0$, which means that a unit increase in $A$ results to a unit decrease of $R_0$. Conversely, as $A$ is just the inverse of the mean residence time in $E$ ($A^{-1}$), a unit increase/decrease in $A^{-1}$ would contribute to a unit increase/decrease in $R_0$, i.e., the faster the individual transitions out of the $E$ compartment (smaller residence time), the more manageable the spread of the pathogen will be (with a reduced $R_0$). Since each term of $A$ is an outflow rate identified with an intervention, this result lends support to more immediate COVID-19 test results combined with a reasonably shorter quarantine/isolation period (days) for reducing the risk of viral spread. However, due to the biological dynamics of COVID-19, there is a minimum required quarantine/isolation period, which imposes a positive lower bound for practical values of $A$ when isolation is combined with limited testing capacity.

4. The inverse proportionality of $B$ to $R_0$ is corroborated by its elasticity index of $-0.99937$. A unit increase of $B$ would lead to a decrease in $R_0$ by $0.99937$ units. Conversely, a unit increase in the mean residence time in the $I_M$ compartment ($B^{-1}$) would lead to an increase of $0.99937$ units to $R_0$. This signifies that the more number of asymptomatic and, or mildly symptomatic COVID-19 cases, the more the pandemic can be controlled. Hence, measures in ensuring that the vulnerable population gets due protection and medical attention produce positive results in preventing the further spread of the virus and disease-related deaths [51, 52].

The disease free, and endemic equilibrium points of the model were also solved and were verified by the proof of its local stability. This meant that two scenarios could be expected to come about in the COVID-19 pandemic dependent on certain conditions: (1) the disease would die out, or (2) becomes endemic [48]. These conditions were then verified to Davao City's circumstances, for which it was found that endemicity would, at best, be the more

favorable scenario to be expected, as the disease free scenario is relatively too difficult to achieve, and at the point when this paper is written, is not yet realistic. Hence, policies toward the management of COVID-19 should consider an endemic scenario where the disease continues to exist but on a controlled manner. Recommendations relative to this are further expounded in the Conclusion part of this paper.

## Sensitivity analysis

The sensitivity analysis helped ascertain which parameters were to be identified or which parameters were to be fixed. Fig 2 presents the LHS/PRCC analysis results for the different response functions considered with the corresponding values presented in Table 4. It was observed that the parameters $\beta$ and $r$ were strongly positively correlated to $I_H$ and $I_M$ cases. In addition to $\beta$ and $r$ parameters, $\delta_M$ was also strongly positively correlated to $I_M$ cases. These indicated a strong association between high values of the $\beta$, $r$, $\delta_M$ parameters to the number of COVID-19 cases, specifically during the early stages of the pandemic. The correlation of these parameters to the response functions decreased over time during the course of the pandemic. This result supported the need to distinguish these three (3) parameters as essential factors in the transmission dynamics of the virus. However, as $\delta_M$ is dependent on the biology of SARS-CoV-2, intervention recommendations focused mainly on policy-preventable parameters $\beta$ and $r$. Moreover, as the severity proportion $q$ and disease-induced death rate $\mu_H$ directly equates to COVID-19 morbidities that we also put attention to these parameters. The implementation of control interventions were essential for the reduction of the values of these parameters ($\beta$, $r$, $q$, and $\mu_H$) to mitigate the spread and lessen the hospitalizations due to COVID-19. Logrosa and colleagues corroborate that if the local government prioritized reducing COVID-19 fatality, the COVID-19 pandemic will be manageable, at least in the context of Davao City, Philippines [27].

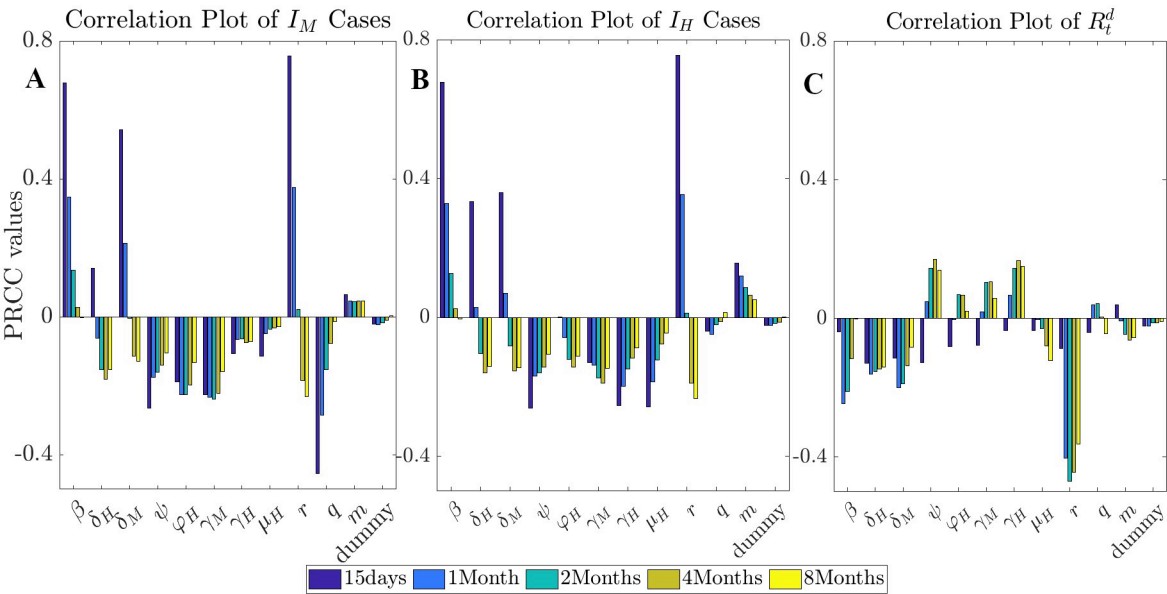

**Fig 2. The PRCC of model parameters.** Using 10,000 LHS samples for the period March 8, 2020, to March 5, 2021, for (A) the number of monitored cases, (B) hospitalized cases, and (C) the deterministic time-varying reproduction number ($R_t^d$). The uncertainty and sensitivity analysis was performed via the LHS/PRCC method.

**Table 4. Output from sensitivity analysis.**

| Parameter | Outcome: $I_M$ | | Outcome: $I_H$ | | Outcome: $R_t^d$ | |
|---|---|---|---|---|---|---|
| | PRCC | p-value | PRCC | p-value | PRCC | p-value |
| $\beta$ | 0.6762* | $1.29e^{-258}$* | *0.6773 | $5.07e^{-240}$* | $-0.2466$ | $2.63e^{-138}$ |
| $r$ | 0.7542** | $3.063e^{-315}$** | 0.7514** | $2.24e^{-274}$** | $-0.4075$ | 0.0000 |
| $\delta_H$ | 0.1605 | $9.12e^{-06}$ | 0.3480 | $9.79e^{-07}$ | $-0.1687$ | $1.09e^{-64}$ |
| $\delta_M$ | 0.5433* | $4.01e^{-110}$* | 0.3710 | $1.04e^{-15}$ | $-0.1878$ | $6.19e^{-80}$ |
| $\gamma_H$ | $-0.1150$ | $1.39e^{-08}$ | $-0.2589$ | $6.42e^{-84}$ | 0.0682 | $8.63e^{-12}$ |
| $\gamma_M$ | $-0.2339$ | $3.52e^{-116}$ | $-0.1417$ | $2.53e^{-36}$ | 0.0370 | 0.0002 |
| $\mu_H$ | $-0.1022$ | $1.53e^{-06}$ | $-0.2479$ | $2.29e^{-79}$ | $-0.0162$ | 0.1052 |
| $q$ | $-0.4346$ | $4.49e^{-164}$ | $-0.0117$ | 0.0034 | 0.0327 | 0.0011 |
| $m$ | 0.0563 | 0.0638 | 0.1529 | $8.88e^{-22}$ | $-0.0454$ | $5.68e^{-06}$ |
| $\psi$ | $-0.2396$ | $9.31e^{-72}$ | $-0.2342$ | $1.08e^{-66}$ | 0.0191 | 0.0569 |
| $\varphi_H$ | $-0.1688$ | $3.59e^{-92}$ | 0.0245 | 0.0033 | $-0.0022$ | 0.8234 |
| dummy | $-0.0002$ | 0.9891 | $-0.0045$ | 0.6349 | $-0.0044$ | 0.6573 |

*possible contributors to uncertainty;

** very likely contributors to uncertainty

## Epidemiological parameters

From the gathered data, Fig 3 shows the COVID-19 daily and cumulative incidence in Davao City over the date of illness onset and the four-time periods that cover the community quarantine levels from March 8, 2020, to March 5, 2021. The complete list of system parameters as a result of the data fitting is presented in Table 3. Fig 3 showed the observed COVID-19 cases and the best model fit to the data and that the model was able to approximate the dynamics of

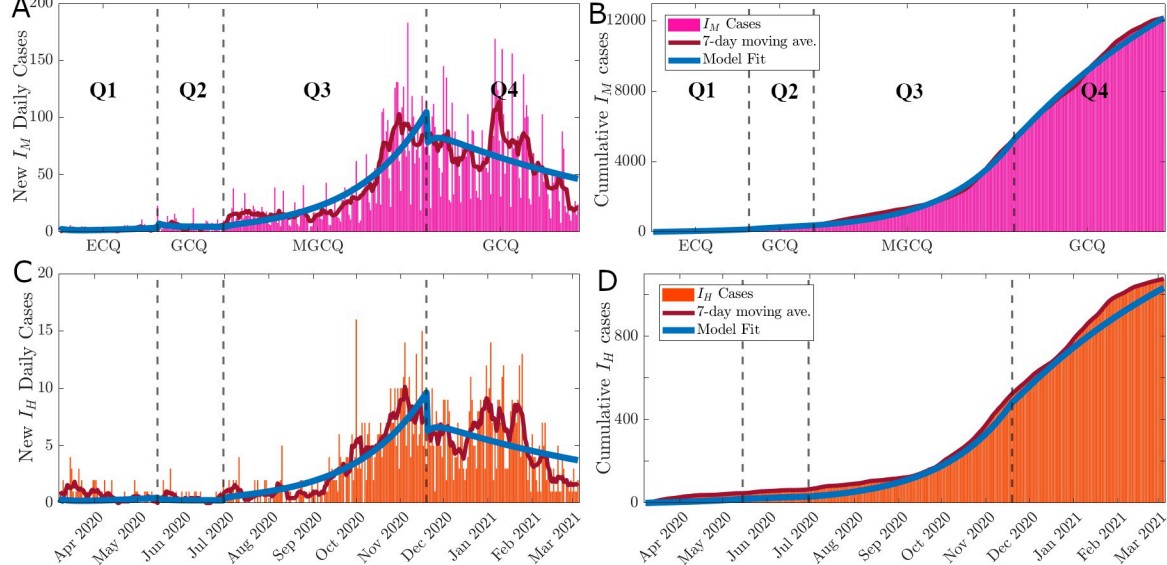

**Fig 3. The best fit of the $SEI_HI_MR$ model to the daily new cases according to health status upon admission.** The four quarantine periods are separated by broken lines. The bars are the daily new incidence data in Davao City, while the solid blue line corresponds to the model simulations. It shows the model fit vs daily data (A), the model fit vs. cumulative data for monitored cases (B), the model fit vs. daily data (C), and the model fit vs. cumulative data for hospitalized cases (D).

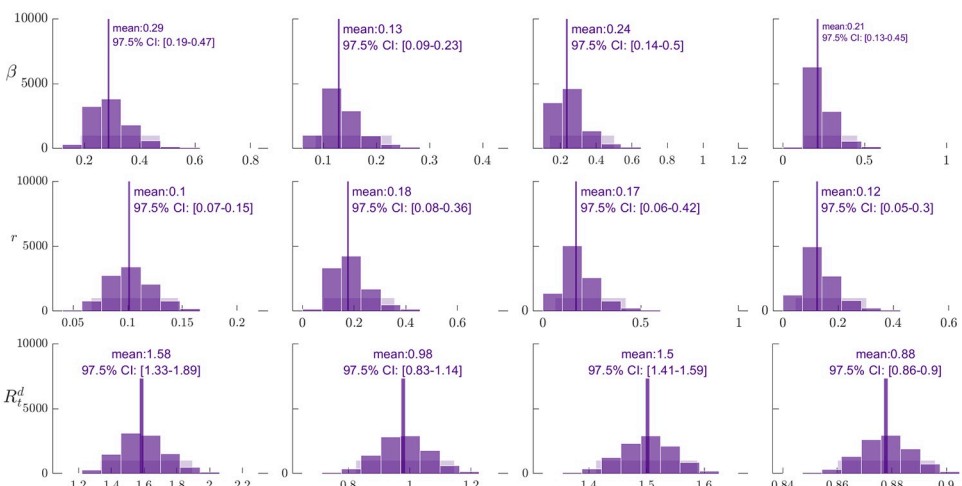

**Fig 4. The histograms obtained from the parametric bootstrapping approach.** These show the empirical distributions of the estimated parameters $\beta$ and $r$, including $R_0$ for (from up to down) Q1, Q2, Q3, and Q4 estimations, respectively. We characterized the empirical distributions using 10,000 Bootstrap realizations assuming a Poisson error structure.

the pathogen. Following the bootstrapping method, the resulting empirical distribution of parameters $\beta$ and $r$, including $R_t^d$ and the 95% confidence intervals of the estimated parameter values are presented in Fig 4. These show that the distribution is normal and that the model and the best fit values were able to capture the transmission dynamics of COVID-19 in Davao City, Philippines.

It was observed in the Figs 3 and 4 that the estimated values of the transmission rate ($\beta$) were at its highest at the start of the pandemic and increased as the quarantine lockdowns were eased over time (compare Table 3). The values show that one infected individual transmits the disease every approximately $\frac{1}{\beta_{Q1}} = \frac{1}{0.2862} \approx 3.49$ days (83.86 hours) during the Q1 period, 7.79 days (186.92 hours) during Q2, 4.24 days (101.74 hours) during Q3, and 4.67 days (112.05 hours) during Q4. The effectiveness of the implemented CQ conform to a perceived lag effect [53], where the strictest measure implemented in ECQ (Q1) led to the least transmission rate at the next classification (Q2). However, since right after Q1 (ECQ; strictest), Q2 (GCQ; most permissive) was implemented, its perceived lag effect brought about a higher transmission rate in Q3, which was almost twice as much as in Q2. The abrupt relaxation of quarantine classification from the strictest to the most relaxed one led to high transmissibility of the disease as more people were allowed mobility. Hence, it would have been best if a gradual relaxation of protocols were followed instead. Nevertheless, comparatively, these estimated values are similar and or within the range of published estimates throughout the world [22], for example, in India [51], and China [52]. Hence, the same as the case of China, with proper stringent preventive measures, the COVID-19 cases in Davao can still be brought down to manageable levels.

Table 3 show that the model estimates of the positivity proportion $r$ were lowest during Q1 and highest during Q2. The results mean that 89.87% of the identified "close contacts" did not develop the disease during Q1, 82.42% during Q2, 82.98% during Q3, and 87.62% during Q4. As Davao City had strict mobility restrictions during Q1 may have led to lower contact among individuals and may have lessen the transmission of the pathogen, thus the tracing and testing capacities were more efficient hence the lowest $r$ value. However, as lockdowns were eased in Q2 and Q3, we observed increases in $r$ despite the tracing and testing efforts. This is due to the

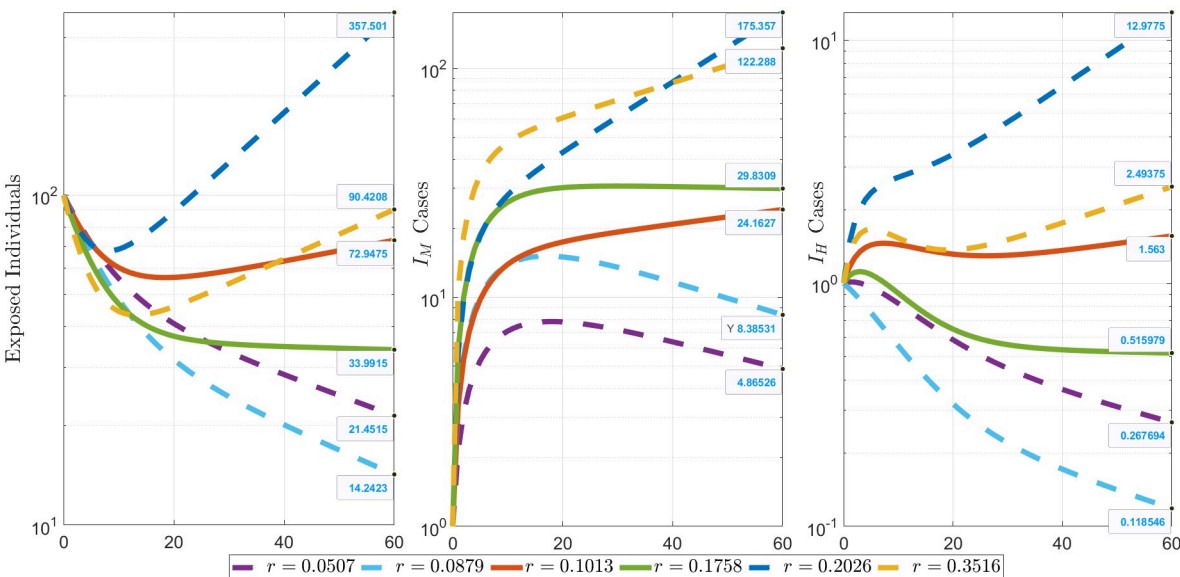

**Fig 5. The different simulations of *r* values and its effect on the transmission dynamics in logscale.** These show the epidemic curves using the smallest estimated *r* = 0.1013, biggest estimated *r* = 0.1758, and scenarios where the *r* values are half (*r* = 0.0507, *r* = 0.0879), and twice the corresponding estimates (*r* = 0.2026, *r* = 0.3516). Solid curves are the simulations using *r* values from the model estimates, while the dashed curves are simulations using arbitrarily chosen *r* values for scenario analysis.

increase of the interacting population size while the tracing and testing capacity remained constant. Furthermore, the testing and tracing guidelines during Q2 and Q3 were a bit more relaxed causing a fewer identified "close contacts", hence the increase of *r*. The decrease of these values would have been a favorable outcome towards controlling the spread of the disease [27].

Different scenarios were considered concerning the different *r* values estimated as shown in Fig 5. It shows that if mass testing was done to at least halve the positivity rate, the daily new exposed population would have reduced from 73 to 21 individuals after 60 days of Q1 and from 34 to 14 individuals during Q2; the daily new $I_M$ cases would have reduced from 24 to 5 individuals after 60 days of Q1 and from 30 to 8 individuals during Q2; and the daily new $I_H$ cases would have become 0 from 2 individuals after 60 days of Q1 and 0 indv from 1 individuals during Q2. Moreover, if the efforts were a bit laxer to twice than what was the positivity rate, the daily new exposed population would have increased to 358 individuals after 60 days of Q1 and to 90 individuals during Q2, the daily new $I_M$ cases would have increased to 175 individuals after 60 days of Q1 and to 122 individuals during Q2, while the daily new $I_H$ cases would have become 13 individuals after 60 days of Q1 and 3 individuals during Q2. These results corroborate with the need for mass testing, contact tracing, and the need for faster release of test results towards mitigating the disease [54].

With respect to the severity proportion *q* as shown in Table 3, we found that 77.29% (1 −*q* = 0.7729) of the confirmed cases are asymptomatic to mild, while 22.71% (*q* = 0.2271) were moderate to critical cases during Q1. Likewise, during Q2, 92.12% are asymptomatic to mild cases, and 7.88% are moderate to critical cases. The moderate to critical cases during Q3 slightly increased with 8.42% of the confirmed cases and 7.42% moderate to critical cases during Q4. The majority of the confirmed cases in Davao City consisted of asymptomatic to mild cases. However, even though the ratio of moderate to critical cases in Davao to asymptomatic to mild cases is relatively low, its total number of cases still amounts to more than Davao City's

total COVID-19 bed capacity. Hence, Davao COVID-19 referral hospitals are usually maxed out of their capacities [55]. Policies and budgetary support towards the increase of Davao City's health system and increasing hospital capacities should be prioritized.

Table 3 further show that in Davao City, a COVID-19-induced death (represented by $\mu_H$) occurs every approximately $\frac{1}{\mu_H} = 8.17$ days in Q1, 6.33 days in Q2, 3.42 days in Q3, and 3.33 days in Q4. Meanwhile, based on the data, the hospitalized cases are discharged at the rate ($\gamma_H$) of 21.5 days in Q1, 20.5 days in Q2, 17 days in Q3, and 20 days in Q4. The infectious period of moderate-critical cases shortens during Q3 as the number of hospitalized individuals eventually reached hospital capacity, which, in turn led to a faster COVID-19-induced death rate [55].

As seen in Fig 6, the overall trend of the deterministic reproduction number $R_t^d$ follows that of the statistical reproduction number $R_t^s$. Furthermore, there is a decrease in the time-varying reproduction number during the Q2 period as compared to Q1. This is during the early stages of the pandemic and the community quarantine intervention has shown to have a positive effect in mitigating the spread of the disease. However, when the government attempted to ease the economic burden by implementing the MGCQ in Q3, the $R_t^d$ increased indicating a faster disease spread in the city. The reimplementation of GCQ in Q4 has reduced the $R_t^d$ to a value of slightly less than one.

It is noteworthy however that at Q3, a difference between the $R_t^d$ and $R_t^s$ can be seen. As Q3 started, the initial conditions of the transmission dynamics were during a spike of cases yielding twice as much $\beta$ value as Q2, hence as discussed in the results of elasticity indices, the spike in $\beta$ also doubled the $R_t^d$. This implies that during the first data points in Q3, $R_t^d$ captured the spike and the extreme values of $R_t^s$. Nevertheless, since $R_t^s$ is data and time-dependent, its values decreased several days after day one of Q3 due to the perceived lag effect of the intervention implemented in that period. From here, it is conjectured that discrepancies between $R_t^d$ and $R_t^s$ in Q3 can be due to the assumed initial conditions (i.e., condition at day 1 of Q3) of the model. Nonetheless, the discrepancies between $R_t^d$ and $R_t^s$ are deemed tolerable.

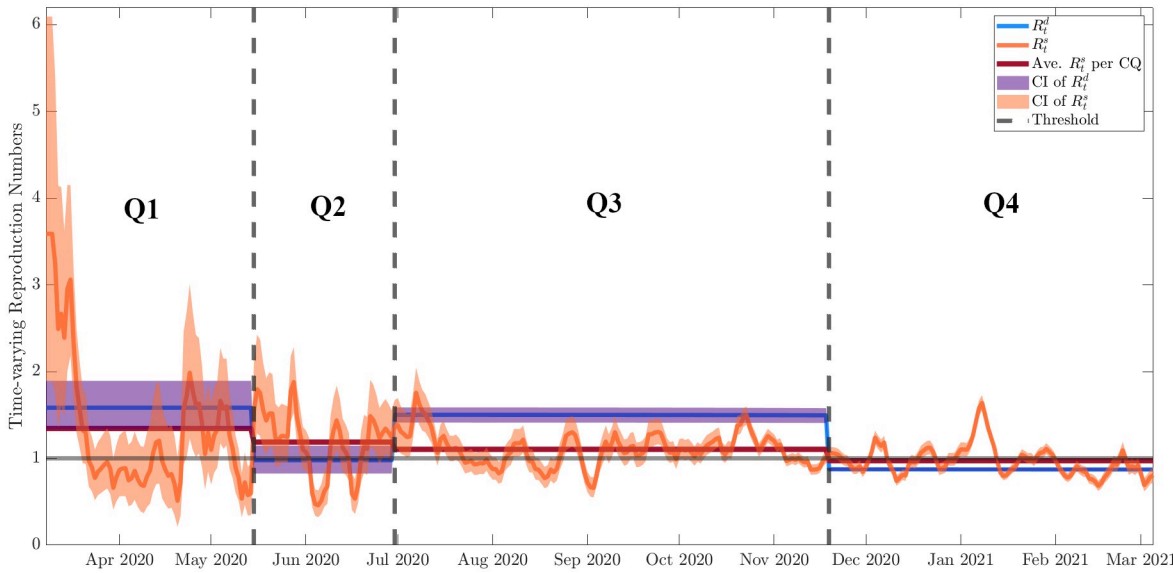

**Fig 6. The deterministic reproduction number $R_t^d$ from the $SEI_HI_MR$ model against the statistical reproduction number $R_t^s$ plot for the time periods considered.**

## Conclusion

This paper presented a community-level COVID-19 transmission model incorporating the country's health department quarantine and isolation protocols with Davao City, Philippines, as a case study. Key epidemiological parameters were represented and were analyzed in conjunction with the different levels and quarantine classifications (e.g., ECQ, GCQ, and MGCQ). Qualitative analyses indicate that the formulated mathematical model is well-posed, bounded, and locally asymptotically stable when it is disease-free, and when the disease is endemic. Using nonlinear least squares techniques, the parameters $\beta$ and $r$ were estimated which were used along with other data-estimated parameters to compute epidemiological measures such as the basic reproduction number, and time-varying reproduction number. Uncertainty and sensitivity analyses were also performed via the LHS/PRCC method.

The result on the reproduction numbers ($R_0$ and $R_t^d$) and elasticity indices as discussed in the Qualitative analyses subsection implied that the number of secondary infections was driven by the number of infectious populations and how often they interact with the susceptible population. The transmission rate in Davao City was estimated to be highest during ECQ, i.e., at the start of the pandemic (Table 3). This indicated that Davao City may had limited and insufficient resources against the emergence of a pandemic. Hence, there is a need to better equip the healthcare system and the local government units (LGU) for better disaster management and prevention of future pandemics. A multi-sectoral framework, policies, and protocols of how the different agencies could collaboratively work should already be put in place even before the occurrence of any health emergencies. The National, Regional, Provincial, City, and Municipal Disaster Risk Reduction and Management Offices, the Department of Health (DOH), the Department of Interior and Local Government, the LGU, Provincial, City, Municipal, and Barangay Health Offices are some of the core agencies necessary for effective pandemic management. Guides, manuals, and protocols should be collaboratively drafted by these agencies and periodic capacity training for its personnel should be incorporated into its mandate for better preparedness and mitigation. Based on the model results, policies toward the reduction of the transmission rate such as the implementation of containment measures, availability and wearing of protective equipment, and sanitation are some of the key factors to consider.

On the other hand, as discussed in the above subsection on Epidemiological parameters and corroborated by Table 3 concerning the transmission rates, an abrupt change from a strict containment measure to a lax one could negatively affect the control measures implemented especially because of a perceived lag effect. A protocol for a gradual easement of containment measures should be implemented instead. Moreover, since we know that the results with respect to the positivity proportion $r$ corroborated the need to put forth better monitoring and identification protocols which include, but are not limited to, testing and contact tracing capabilities, and travel and mobility surveillance. Furthermore, the results for the severity proportion $q$ suggested the need to allocate funding and to study the appropriate ratio of the hospital carrying capacity to the city population because despite the small number of severely symptomatic COVID-19 patients, the hospitals in Davao City still were overwhelmed. More than allocating funds for equipment and buildings, the production of and adequate remuneration for an appropriate number of nursing and healthcare professionals should be a necessity [56]. The need for this action was further supported by the results with respect to the disease-induced death rate $\mu_H$ as it only critically increased in Q4 when many of the hospitals reached its maximum capacity, and many of the healthcare workers were overwhelmed by the number of hospital admissions. Despite having available hospital beds, the number of attending nurses and doctors was inadequate, and the available ones are already highly exhausted. Government

interventions such as the provisions of scholarships, opening of job positions, and offering of competitive remuneration among others should have more investments and be made more accessible to all aspiring healthcare workers.

Overall, the outputs generated from this study are potentially beneficial as a basis in making decisions crucial for impeding the spread of COVID-19 and may be a baseline basis for the protocols against future pandemics. These outputs also provided a quantitative measure of the respective effect of the various measures implemented during the different quarantine classifications. This model could be used for other cities and regions in the Philippines to assess the effects of their respective efforts to combat the disease, which is contextualized on a community level. Moreover, researchers and practitioners must be aware of the limitations of compartmental-based modeling (SIR, SEIR, etc.). This approach generally assumes that the community under infection is homogeneous: each human host infects and undergoes infection in the same manner. A comparison of our analysis with agent-based models [57] and other modelling techniques accounting heterogeneity can reveal additional factors that can refine epidemic dynamics and projections. Since our model assumed data prior to the vaccine rollout in Davao City, a suitable refinement of our model should incorporate community-wide inoculation dynamics. Other possible extensions account for risk groups [58], particularly age groups [59], and untraced and undetected infectious individuals in the context of testing and contact tracing [54, 60].

## Supporting information

**S1 File. Supplemental file.** This file contains solutions to the mathematical theories described in this paper.
(PDF)

## Acknowledgments

We acknowledge Honey Glenn P. Lorono for her technical writing assistance in making this paper easily readable, the DOH Davao Center for Health Development RESU for sharing their data with us, and the Interdisciplinary Applied Modeling team in UP Mindanao and Federico M. Calo for assisting us in the data handling. ADLRV acknowledges the support of the Institute for Basic Science (IBS-R029-C3).

## Author Contributions

**Conceptualization:** Zython Paul T. Lachica, Alexis Erich S. Almocera, Jayrold P. Arcede, May Anne E. Mata.

**Data curation:** Zython Paul T. Lachica, Jayve Iay E. Lato, Jolly Mae G. Catalan, Rachel Joy F. Pasion, Annabelle P. Yumang, May Anne E. Mata.

**Formal analysis:** Loreniel E. Añonuevo, Zython Paul T. Lachica, Deza A. Amistas, Jayve Iay E. Lato, Hanna Lyka C. Bontilao, Alexis Erich S. Almocera, Jayrold P. Arcede, May Anne E. Mata, Aurelio A. de los Reyes V.

**Funding acquisition:** Zython Paul T. Lachica, Alexis Erich S. Almocera, Jayrold P. Arcede.

**Investigation:** Loreniel E. Añonuevo, Zython Paul T. Lachica, Deza A. Amistas, Jayve Iay E. Lato, Hanna Lyka C. Bontilao, Jolly Mae G. Catalan.

**Methodology:** Loreniel E. Añonuevo, Zython Paul T. Lachica, Jayve Iay E. Lato, Hanna Lyka C. Bontilao, Alexis Erich S. Almocera, Jayrold P. Arcede, May Anne E. Mata, Aurelio A. de los Reyes V.

**Project administration:** Zython Paul T. Lachica, Jayrold P. Arcede, May Anne E. Mata.

**Resources:** Jayve Iay E. Lato, Jolly Mae G. Catalan, Rachel Joy F. Pasion, Annabelle P. Yumang, Alexis Erich S. Almocera, Jayrold P. Arcede, May Anne E. Mata, Aurelio A. de los Reyes V.

**Software:** Loreniel E. Añonuevo, Deza A. Amistas, Hanna Lyka C. Bontilao, May Anne E. Mata, Aurelio A. de los Reyes V.

**Supervision:** Alexis Erich S. Almocera, Jayrold P. Arcede, May Anne E. Mata, Aurelio A. de los Reyes V.

**Validation:** Jayve Iay E. Lato, Jolly Mae G. Catalan, Rachel Joy F. Pasion, Annabelle P. Yumang, Alexis Erich S. Almocera, May Anne E. Mata, Aurelio A. de los Reyes V.

**Visualization:** Loreniel E. Añonuevo, Deza A. Amistas, Jayve Iay E. Lato, Hanna Lyka C. Bontilao, May Anne E. Mata, Aurelio A. de los Reyes V.

**Writing – original draft:** Loreniel E. Añonuevo, Zython Paul T. Lachica, Deza A. Amistas, Jayve Iay E. Lato, Hanna Lyka C. Bontilao, Alexis Erich S. Almocera, Jayrold P. Arcede, May Anne E. Mata, Aurelio A. de los Reyes V.

**Writing – review & editing:** Loreniel E. Añonuevo, Zython Paul T. Lachica, Deza A. Amistas, Jayve Iay E. Lato, Jolly Mae G. Catalan, Alexis Erich S. Almocera, Jayrold P. Arcede, May Anne E. Mata, Aurelio A. de los Reyes V.

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
