## [Decision Letter · Decision Letter 0]

5 Sep 2022

PONE-D-22-14208Transmission dynamics and baseline epidemiological parameter estimates of Coronavirus disease 2019 (COVID-19) pre-vaccination: Davao City, PhilippinesPLOS ONE

Dear Dr. Mata,

Thank you for submitting your manuscript to PLOS ONE. After careful consideration, we feel that it has merit but does not fully meet PLOS ONE’s publication criteria as it currently stands. Therefore, we invite you to submit a revised version of the manuscript that addresses the points raised during the review process.

We look forward to receiving your revised manuscript.

Kind regards,

Eunok Jung, Ph.D.

Academic Editor

PLOS ONE

Journal Requirements:

2. Please note that PLOS ONE has specific guidelines on code sharing for submissions in which author-generated code underpins the findings in the manuscript. In these cases, all author-generated code must be made available without restrictions upon publication of the work. Please review our guidelines at https://journals.plos.org/plosone/s/materials-and-software-sharing#loc-sharing-code and ensure that your code is shared in a way that follows best practice and facilitates reproducibility and reuse

"This work is being supported by the Department of Science and Technology-Philippine Council for Health Research and Development (DOST-PCHRD), through the PPASTOL research project under the Niche Center in the Regions for R&D (NICER) Program on Decision Support Systems in Health based in the University of the Philippines Mindanao. We also acknowledged Honey Glenn P. Lorono for her technical writing assistance in making this paper easily readable, the DOH Davao Center for Health Development RESU for sharing their data to us, and the Interdisciplinary Applied Modeling team in UP Mindanao for assisting us in the data handling. 

"This research is funded by the Department of Science and Technology-Philippine Council for Health Research and Development through its Niche Center in the Regions for Research and Development, "Center for Applied Modeling, Data Analytics, and Bioinformatics for Decision Support Systems in Health"."

Reviewers' comments:

Reviewer's Responses to Questions

**Comments to the Author**

1. Is the manuscript technically sound, and do the data support the conclusions?

Reviewer #1: Yes

Reviewer #2: Partly

2. Has the statistical analysis been performed appropriately and rigorously? 

Reviewer #1: No

Reviewer #2: Yes

3. Have the authors made all data underlying the findings in their manuscript fully available?

Reviewer #1: Yes

Reviewer #2: No

4. Is the manuscript presented in an intelligible fashion and written in standard English?

Reviewer #1: Yes

Reviewer #2: No

5. Review Comments to the Author

Reviewer #1: The manuscript is interesting and relevant. However, some major and minor issues have to be addressed first before the manuscript can be accepted for publication in PLOS One.

Major Comments

1. Lines 107 and 116 are contradictory. How can I_H and I_M have close contact with S if they are isolated? How about considering another compartment (Q), which denotes individuals who are isolated and not infectious?

2. One of the main contributions of the paper is introducing the parameter r. This is interesting and realistic given the situation in Davao. I recommend that the author perform more simulations on this. For example, do some scenarios for varying r and see the effects on the number of cases. This might add merit to the work.

3. The figures are poorly generated. Some of the legends are not readable.

4. Lines 210-211 and Table 1 mention that some of the parameters were statistically solved from the data and a reference [37] was cited. However, this reference is not available online and seems like a study made by the research group. I suggest that the authors add a supplementary file detailing how each of the statistically derived parameters is calculated.

5. Lines 211-212 mention that some of the parameters are obtained from existing published literature? Which parameters? Which references? The authors may add another column in Table 1 to indicate the sources of the parameters.

6. Please comment on a significant difference between the statistical R_t and the deterministic R_t from July 1 to the middle of December.

Minor Comments

1. Figure 2 may be omitted. I think Figure 4 is enough.

2. I suggest using 10,000 samples in LHS-PRCC and 10,000 realizations in the bootstrapping.

3. Incorrect citation in line 186. There was no mention of LHS-PRCC in the works of Massey and Colleagues [42]. The authors may also add the link to the GitHub site.

4. Figures 6 and 7 may be omitted. I think Figure 5 is enough.

5. Add a year on the xtick labels of Figure 8.

Reviewer #2: The reviewer report has been uploaded as a word document.

6. PLOS authors have the option to publish the peer review history of their article (what does this mean?). If published, this will include your full peer review and any attached files.

Reviewer #1: No

Reviewer #2: No

---

## [Author Response · Author response to Decision Letter 0]

6 Dec 2022

Please refer to the Response to Reviewer file for the detailed response of each comment. The cover letter has addressed as well the editor's comments. Thank you.

---

## [Decision Letter · Decision Letter 1]

2 Mar 2023

Transmission dynamics, baseline epidemiological parameter estimates of Coronavirus disease 2019 pre-vaccination: Davao City, Philippines

PONE-D-22-14208R1

Dear Dr. Mata,

We’re pleased to inform you that your manuscript has been judged scientifically suitable for publication and will be formally accepted for publication once it meets all outstanding technical requirements.

Kind regards,

Eunok Jung, Ph.D.

Academic Editor

PLOS ONE

Additional Editor Comments (optional):

I am more than happy to accept the article. However, please revise the abstract because it is too general. Highlight the important finding and originity of your study.

Reviewers' comments:

Reviewer's Responses to Questions

**Comments to the Author**

1. If the authors have adequately addressed your comments raised in a previous round of review and you feel that this manuscript is now acceptable for publication, you may indicate that here to bypass the “Comments to the Author” section, enter your conflict of interest statement in the “Confidential to Editor” section, and submit your "Accept" recommendation.

Reviewer #1: All comments have been addressed

2. Is the manuscript technically sound, and do the data support the conclusions?

Reviewer #1: Yes

3. Has the statistical analysis been performed appropriately and rigorously? 

Reviewer #1: Yes

4. Have the authors made all data underlying the findings in their manuscript fully available?

Reviewer #1: Yes

5. Is the manuscript presented in an intelligible fashion and written in standard English?

Reviewer #1: Yes

6. Review Comments to the Author

Reviewer #1: The authors have addressed all my comments and recommendations. The paper can be accepted in its current form.

7. PLOS authors have the option to publish the peer review history of their article (what does this mean?). If published, this will include your full peer review and any attached files.

Reviewer #1: No

---

## [Editor Report · Acceptance letter]

29 Mar 2023

PONE-D-22-14208R1 

Transmission dynamics and baseline epidemiological parameter estimates of Coronavirus disease 2019 pre-vaccination: Davao City, Philippines 

Dear Dr. Mata:

I'm pleased to inform you that your manuscript has been deemed suitable for publication in PLOS ONE. Congratulations! Your manuscript is now with our production department. 

Kind regards, 

on behalf of

Dr. Eunok Jung 

Academic Editor

PLOS ONE